# Nirmatrelvir-resistant SARS-CoV-2 is efficiently transmitted in female Syrian hamsters and retains partial susceptibility to treatment

Rana Abdelnabi [1,2], Dirk Jochmans [1], Kim Donckers[1], Bettina Trüeb[3,4], Nadine Ebert[3,4], Birgit Weynand [5], Volker Thiel [3,4] & Johan Neyts [1,2,6] ✉

The SARS-CoV-2 main protease (3CLpro) is one of the promising therapeutic targets for the treatment of COVID-19. Nirmatrelvir is the first 3CLpro inhibitor authorized for treatment of COVID-19 patients at high risk of hospitalization. We recently reported on the in vitro selection of SARS-CoV-2 3CLpro resistant virus (L50F-E166A-L167F; 3CLpro[res]) that is cross-resistant with nirmatrelvir and other 3CLpro inhibitors. Here, we demonstrate that the 3CLpro[res] virus replicates efficiently in the lungs of intranasally infected female Syrian hamsters and causes lung pathology comparable to that caused by the WT virus. Moreover, hamsters infected with 3CLpro[res] virus transmit the virus efficiently to co-housed non-infected contact hamsters. Importantly, at a dose of 200 mg/kg (BID) of nirmatrelvir, the compound was still able to reduce the lung infectious virus titers of 3CLpro[res]-infected hamsters by 1.4 $\log_{10}$ with a modest improvement in the lung histopathology as compared to the vehicle control. Fortunately, resistance to Nirmatrelvir does not readily develop in clinical setting. Yet, as we demonstrate, in case drug-resistant viruses emerge, they may spread easily which may thus impact therapeutic options. Therefore, the use of 3CLpro inhibitors in combination with other drugs may be considered, especially in immunodeficient patients, to avoid the development of drug-resistant viruses.

The Severe Acute Respiratory Syndrome coronavirus 2 (SARS-CoV-2), the causative agent of COVID-19, has had a devastating impact on global public health since its emergence in Wuhan (China) in December 2019. So far, three antiviral drugs have been approved/authorized by FDA and EMA for clinical use in COVID-19 patients i.e. the nucleoside analogs remdesivir and molnupiravir and the viral protease (3CLpro/Mpro) inhibitor nirmatrelvir[1].

SARS-CoV-2 3CLpro is a cysteine protease which cleaves the two viral polyproteins (pp1a and pp1ab) at eleven different sites, resulting in various non-structural proteins, which are essential for viral

[1]KU Leuven Department of Microbiology, Immunology and Transplantation, Rega Institute for Medical Research, Laboratory of Virology and Chemotherapy, B-3000 Leuven, Belgium. [2]The VirusBank Platform, Gaston Geenslaan, B-3000 Leuven, Belgium. [3]Institute of Virology and Immunology, University of Bern, 3012 Bern, Switzerland. [4]Department of Infectious Diseases and Pathobiology, Vetsuisse Faculty, University of Bern, Bern, Switzerland. [5]KU Leuven Department of Imaging and Pathology, Division of Translational Cell and Tissue Research, B-3000 Leuven, Belgium. [6]Global Virus Network, GVN, Baltimore, US. ✉e-mail: johan.neyts@kuleuven.be

replication[2,3]. Interestingly, the cleavage site of the SARS-CoV-2 3CLpro substrate is not recognized by any known human proteases[4,5]. Therefore, 3CLpro is a selective antiviral drug target. Nirmatrelvir (PF-07321332) is a peptidomimetic reversible covalent inhibitor of 3CLpro that is co-formulated with the pharmacokinetic enhancer ritonavir (the resulting combination being marketed as Paxlovid)[6]. When treatment is initiated during the first days after symptom onset, it results in roughly 90% protection against severe COVID-19 and hospitalization[7]. Besides nirmatrelvir, the non-peptidic, non-covalent inhibitor, ensitrelvir (S-217622, Xocova®) has been approved in Japan for emergency use against SARS-CoV-2[8]. Other 3CLpro inhibitors are also currently in clinical development.

The emergence of drug-resistant viruses is a major concern when using antivirals. Development of drug resistant variants has been reported during antiviral treatment against different infections including with the human immunodeficiency virus (HIV), hepatitis B virus (HBV), hepatitis C virus (HCV), herpesviruses and influenza virus[9–12]. Moreover, transmission of drug resistant viruses has also been reported for HIV[13] and influenza virus[14]. Selection of remdesivir-resistant SARS-CoV-2 variants has been also reported in cell culture and in clinical settings[15–17]. On the other hand, there are no clear data available yet about emergence of resistant virus variants to molnupiravir or nirmatrelvir in treated patients. Recently, we reported on in vitro selection of a SARS-CoV-2 resistant virus against a first generation 3CLpro inhibitor (ALG-097161) that is cross-resistant to several 3CLpro inhibitors including nirmatrelvir[18]. The identified resistant virus carries three amino acid substitutions in 3CLpro (L50F-E166A-L167F) that result in a more than 20 fold increase in $EC_{50}$ values for different 3CLpro-inhibitors[18]. These substitutions are associated with a significant loss of the 3CLpro activity in enzymatic assays, suggesting a subsequent reduction in viral fitness[18].

Here, we aimed to (i) explore the infectivity and virulence of the in vitro selected 3CLpro (L50F-E166A-L167F) nirmatrelvir resistant (3CLpro$^{res}$) virus in Syrian hamsters, (ii) assess the transmission potential of this in vitro selected drug resistant virus from intranasally infected index hamsters to non-infected contact hamsters and (iii) evaluate the efficacy of nirmatrelvir against this resistant variant in the hamster infection model.

## Results

To assess the infectivity and the transmission potential of the 3CLpro (L50F-E166A-L167F) nirmatrelvir-resistant (3CLpro$^{res}$) virus in animals, two groups of index hamsters (each $n = 12$) were intranasally infected with $1 \times 10^4$ TCID$_{50}$ of either the wild-type (WT) SARS-CoV-2 virus (USA-WA1/2020) or the reverse engineered nirmatrelvir-resistant virus carrying substitutions L50F-E166A-L167F in 3CLpro[18]. On day 1 post-infection (pi), each of the index hamsters was co-housed in a cage with a contact hamster. The co-housing continued until 3 days after start of contact (Fig. 1a). All hamsters were then euthanized (day 4 pi). Index hamsters infected with the WT virus had median viral RNA and infectious virus loads in the lungs of $1.3 \times 10^7$ genome copies/mg tissue (Figs. 1b) and $1.2 \times 10^5$ TCID$_{50}$/mg tissue (Fig. 1c), respectively. The 3CLpro$^{res}$ virus replicated also efficiently in the lungs of infected index hamsters but with lower viral loads than the WT virus. For the 3CLpro$^{res}$ virus, median viral RNA load $= 3.1 \times 10^6$ genome copies/mg lung tissue [Fig. 1b, $p = 0.0009$] and a median infectious virus titers $= 1.6 \times 10^3$ TCID$_{50}$/mg lung tissue [Fig. 1c, $p < 0.0001$], which is respectively 0.6 and 1.9 log$_{10}$ lower than is the case for WT virus.

All sentinels that had been co-housed with either WT or the 3CLpro$^{res}$ virus-infected index hamsters had detectable viral RNA in their lungs in the range of $(6.9 \times 10^3 – 5.7 \times 10^7)$ and $(2.2 \times 10^3 – 1.1 \times 10^7)$ genome copies/mg lung tissue, respectively (Fig. 1b). For contacts co-housed with index hamsters that had been infected with WT virus,

infectious virus titers were detected in the lungs of 11 out of 12 contact animals with infectious titer loads in the lungs ranging from $1.2 \times 10^3 – 2 \times 10^5$ TCID$_{50}$/mg lung tissue (Fig. 1C). On the other hand, 9 out of 12 contacts that had been co-housed with index hamsters infected with the 3CLpro$^{res}$ virus, showed detectable infectious virus titers in their lungs ranging from $2 \times 10^1 – 7.5 \times 10^4$ TCID$_{50}$/mg lung (Fig. 1c).

Histological study of the lungs from the two infected index groups revealed that both the WT and the 3CLpro$^{res}$ virus caused comparable pathological signs including endothelialitis, peri-vascular inflammation, peri-bronchial inflammation and bronchopneumonia (Fig. 2a). Moreover, the median cumulative histopathological lung scores of index hamsters infected with either the WT virus or the 3CLpro$^{res}$ were not different (Fig. 2b). No significant differences in %body weight change were observed between the WT and 3CLpro$^{res}$ virus-infected groups throughout the infection period (supplementary Fig. S1). Deep sequencing analysis of viral RNA isolated from the lungs of index and contact hamsters infected with the 3CLpro$^{res}$ virus revealed that the L50F-E166A-L167F substitutions were maintained in the 3CLpro of this resistant virus after replication in hamsters, indicating the genomic stability of these mutations.

Next, we explored the antiviral efficacy of nirmatrelvir against the WT and the 3CLpro$^{res}$ virus in the hamster model. To that end, animals were treated with either vehicle or nirmatrelvir (at 100 or 200 mg/kg) twice daily by oral gavage for 4 consecutive days, starting just before the infection [with the WT or the 3CLpro$^{res}$ virus (Day 0)]. On day 4 pi, all animals were euthanized and lungs were collected to quantify the infectious virus titers and evaluate the lung pathology (Fig. 3A). Treatment of the WT virus-infected hamsters with nirmatrelvir at 100 or 200 mg/kg (BID) resulted in respectively a 1.2 log$_{10}$ ($P = 0.022$, supplementary Fig. S2A) and 2.2 log$_{10}$ ($P = 0.0003$, Fig. 3b) reduction of infectious virus titers compared to the vehicle group infected with WT virus. In hamsters infected with the the 3CLpro$^{res}$ virus, the 200 mg/kg (BID) resulted in a reduction of viral replication (1.4 log$_{10}$) as compared to the corresponding vehicle group (Fig. 3b), whereas no significant reduction in infectious virus titers was observed in the 100 mg/kg dose group (supplementary Fig. S2A). In addition, a significant reduction ($P < 0.0001$) of lung histopathology scores (close the baseline score of uninfected untreated hamsters) was observed in the nirmatrelvir (200 mg/kg, BID)-treated group infected with the WT virus (Fig. 3C). Treatment of animals infected with the WT virus with 100 mg/kg (BID) of the compound also resulted in a significant improvement of lung histopathology scores (median score of 3.5, p = 0.035) as compared to the corresponding vehicle group (supplementary Fig. S2B). On the other hand, treatment of the hamsters infected with the 3CLpro$^{res}$ virus with nirmatrelvir at 200 mg/kg (BID) resulted in slight (non-significant) reduction of lung histopathology scores (median score of 4.25, Fig. 3c) as compared to the corresponding vehicle group (median score of 5.5, Fig. 3c) and no reduction in pathology scores was observed in the 100 mg/kg (BID) dose group (supplementary Fig. S2B). No significant weight loss was observed in either the vehicle or niramtrelvir-treated groups on day 4 pi (Fig. 3d). However, the WT virus-infected group that was treated with nirmatrelvir had a significantly higher body weight gain (mean % body weight change of 4.1, $P = 0.0058$), as compared to the other groups (mean % body weight change <2) (Fig. 3d). Sequencing of viral RNA obtained from the lungs revealed no changes in the 3CLpro gene of the WT virus as well as no additional mutations in the 3CLpro of the resistant variant (supplementary Fig. S3). Passaging the WT virus through the hamster seems to select for several mutations in the S-protein which is not the case for the 3CLpro$^{res}$ virus (supplementary Fig. S4).

## Discussion

So far SARS-CoV-2 drug resistant variants have only been reported in patients treated with the viral polymerase inhibitor remdesivir[16,19]. We

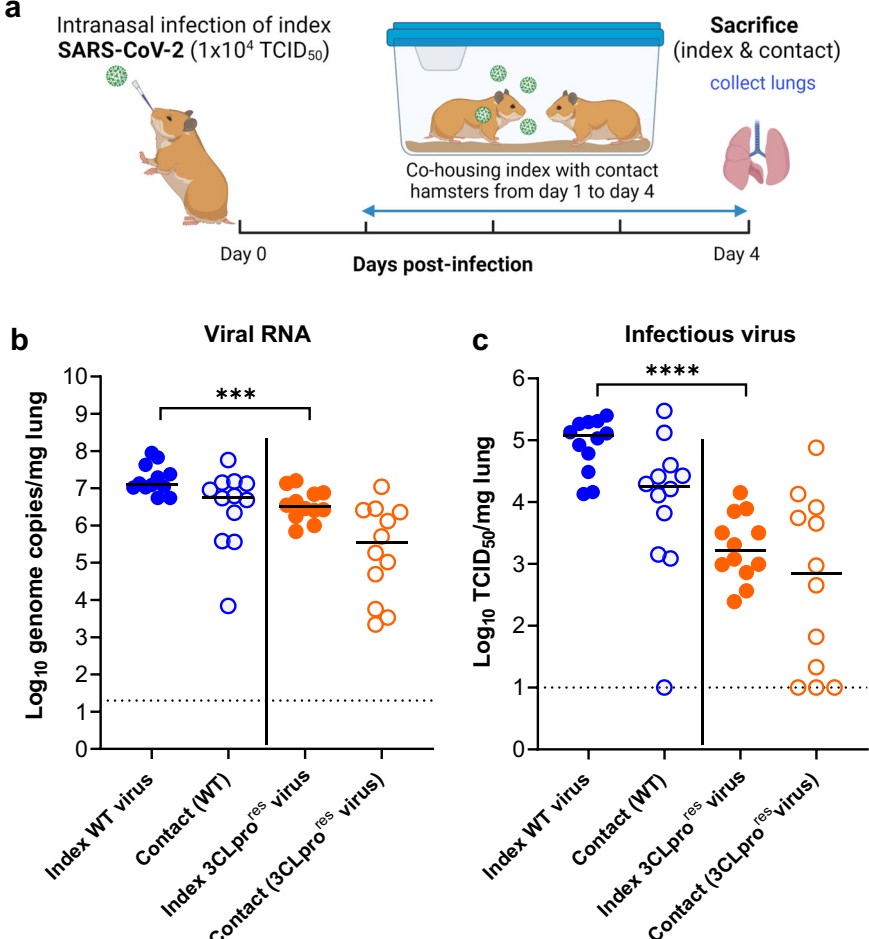

**Fig. 1 | The effect of 3CLpro substitutions, that are associated with nirmatrelvir-resistance, on viral transmission to contact hamsters. a** Set-up of the study. **b** Viral RNA levels in the lungs of index hamsters infected with $10^4$ $TCID_{50}$ of either the wild-type (WT) SARS-CoV-2 isolate (USA-WA1/2020) or the 3CLpro (L50F-E166A-L167F) nirmatrelvir resistant (3CLpro[res]) virus (closed circles) and contact hamsters (open circles) at day 4 post-infection (pi) are expressed as $\log_{10}$ SARS-CoV-2 RNA copies per mg lung tissue. Individual data and median values are

presented. **c** Infectious viral loads in the lungs of SARS-CoV-2 WT and (3CLpro[res]) virus-infected index hamsters (closed circles) and contact hamsters (open circles) at day 4 pi are expressed as $\log_{10}$ $TCID_{50}$ per mg lung tissue. Individual data and median values are presented. Data were analyzed with the two sided Mann–Whitney $U$-test, ***$p = 0.0009$, ****$p < 0.0001$. Data presented are from 2-independent studies with a total $n = 12$ per group. Panel (**a**) designed by Biorender.

have recently reported on the in vitro selection of resistant SARS-CoV-2 against a number of 3CLpro inhibitors including nirmatrelvir and ensitrelvir. This virus carries the L50F-E166A-L167F substitutions in the 3CLpro protein[18]. In a FRET-based assay, the enzymatic activities of recombinant 3CLpro proteins carrying the L50F, E166A and L167F substitutions alone or combined proved to be significantly lower, compared to that of the WT protein[18]. In another similar study, in vitro selection of resistant variants against nirmatrelvir resulted in identification of several resistance-associated substitutions with the E166V substitution resulting in the strongest resistance phenotype[20]. However, this substitution at position 166 resulted in loss of the virus replication fitness in vitro that was restored by compensatory substitutions such as L50F and T21I[20].

Here, we assessed the in vivo fitness of this 3CLpro (L50F-E166A-L167F) nirmatrelvir (and other protease inhibitors) resistant virus in terms of infectivity and transmission potential in a hamster infection model. Intranasal infection of female Syrian hamsters with either the WT or the 3CLpro[res] virus resulted in efficient replication of the virus in the lungs. However, the viral RNA loads and infectious titers in the lungs of hamsters infected with the 3CLprores virus were 0.6 and 1.9 $\log_{10}$ lower compared to the WT virus-infected index hamsters. The relative lower infectious virus titers as compared to the RNA levels of

the 3CLpro[res] virus (isolated from the lungs of hamsters) seems to indicate that this 3CLpro[res] virus has a lower relative infectivity (ratio viral RNA/infectious titers) than the WT virus. We have as yet no explanation of this observation. On the other hand, both the WT and the 3CLpro[res] virus caused a comparable lung pathology in the infected index hamsters.

Co-housing of each index animal, infected with either the WT or the 3CLpro[res] virus, in a cage with a contact (non-infected hamster) for 3 days revealed efficient transmission of either virus between infected and non-infected hamsters. All contact hamsters from both groups had detectable viral RNA loads in their lungs. Infectious virus titers were detected in the lungs of respectively 92% and 75% of the sentinels co-housed with animals that had been infected with either the WT virus or 3CLpro[res] virus. The discrepancy between the number of contact hamsters with viral RNA in the lungs and detectable infectious virus in their lungs might be explained by the fact that all the contact animals were sacrificed at the same day (day 3 post first contact) whereas transmission most likely does not occur in a synchronized way; some animals may possibly soon, others later after contact be infected. Accordingly, for animals that acquired the virus rather late after the first contact, viral RNA might already be detectable

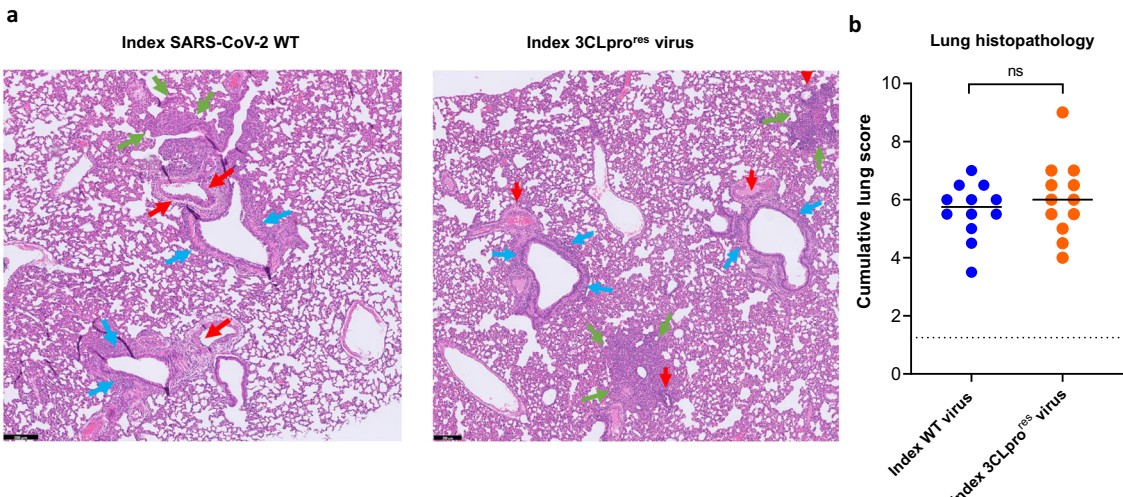

**Fig. 2 | Histopathology of lungs of Syrian hamsters infected with either the wild-type SARS-CoV-2 or the 3CLpro (L50F-E166A-L167F) resistant virus.**
**a** Representative H&E images of lungs of hamsters infected with $10^4$ $TCID_{50}$ of either the wild-type (WT) SARS-CoV-2 virus (USA-WA1/2020) or the 3CLpro (L50F-E166A-L167F) nirmatrelvir resistant (3CLpro[res]) virus at day 4 post-infection (pi) showing peribronchial inflammation (blue arrows), peri-vascular inflammation with vasculitis (red arrows), and foci of bronchopneumonia (green arrows). Scale bars, 200 μm. **b** Cumulative severity score from H&E stained slides of lungs from hamsters infected with the WT virus or the (3CLpro[res]) virus at day 4 pi. Individual data and median values are presented and the dotted line represents the median score of untreated non-infected hamsters. Data were analyzed with the two-sided Mann−Whitney $U$-test, ns = non-significant ($p = 0.5$). Data presented are from 2-independent studies with a total $n = 12$ per group.

in the lungs, but one or two more days may have been required to allow detection of infectious titers.

We also explored the antiviral efficacy of nirmatrelvir against the 3CLpro[res] virus versus the WT virus in the hamster model. Although we observed a marked reduction in the antiviral efficacy of nirmatrelvir against the resistant virus (compared to WT), the compound was still able to reduce the infectious virus titers to some extent (by 1.4 $log_{10}$) at 200 mg/kg dose. A similar pattern of reduced nirmatrelvir efficacy against the drug-resistant virus was also observed in the lung histopathology scores. In an earlier study in which we studied the effect of nirmatrelvir in the SARS-CoV2 hamster infection model, a slightly higher dose of the compound was used (i.e. 250 mg/kg instead of 200 mg/kg in the present study)[21]. This 250 mg/kg dose resulted in peak plasma levels of 22.4 ± 11 μM[21] and peak lung exposure of 6.2 ± 4.9 μM (at 3 h post exposure, unpublished data). In Vero E6, the $EC_{50}$ of nirmatrelvir against the 3CLpro[res] virus is 6.1 μM[18]. With the 200 mg/kg dose (used in the current study), lung exposure may be expected to be somewhat below 6 μM; which may explain the residual activity of this dose against the resistant virus. Modeling based on human pharmacokinetic data of nirmatrelvir will be needed to estimate whether nirmatrelvir treatment may still retain some efficacy in patients carrying such a drug-resistant virus.

Taken together, these results show that the 3CLpro (L50F-E166A-L167F) nirmatrelvir resistant virus is able to efficiently replicate in the lungs of Syrian hamsters and that this virus can be transmitted via direct contact to co-housed naive hamsters. Nirmatrelvir is less efficient in animals infected with the resistant virus than in WT-infected hamsters, but still retains a certain level of antiviral activity. Two of the selected substitutions (i.e. L50F and L167F) have been reported to be detected (at low frequencies) in SARS-CoV-2 viruses naturally circulating in the population (17). Moreover, the Paxlovid label indicates that the E166V subsitition is among the treatment-emergent substitution, which is more common in nirmatrelvir/ritonavir treated subjects relative to placebo-treated subjects. These observations suggest that the in vitro selected 3CLpro (L50F-E166A-L167F) resistant virus may also emerge in treated patients with increased use of Paxlovid or other 3CLpro targeting drugs in the future. The emergence of drug-resistant SARS-CoV-2 variants that can be efficiently transmitted

in the population is of serious concern. It may result in reduced potency of an important therapeutic option for patients that acquire the drug-resistant virus. Efficient transmission of the anti-influenza drugs amantadine and rimantadine has been well documented[14] although it should be mentioned that the barrier to resistance of these amantadines is much lower than that of the SARS-CoV-2 3CLpro inhibitors currently being developed. For the treatment of infections with HIV and HCV fixed-dose combinations prevent the emergence of resistant variants[22]. It will be prudent and important to explore the efficacy and impact on potential resistance development of combinations of SARS-CoV-2 3CLpro inhibitors with drugs with a non-overlapping resistance profile.

## Methods
### SARS-CoV-2
The WT SARS-CoV-2 strain used in this study, SARS-CoV-2 USA-WA1/2020 (EPI_ISL_404895, passaged 3 times on Vero cells then twice on Vero E6 cells), was obtained through BEI Resources (ATCC) cat. No. NR-52281, batch 70036318. This isolate has a close relation with the prototypic Wuhan-Hu-1 2019-nCoV (GenBank accession 112 number MN908947.3) strain as confirmed by phylogenetic analysis. The reverse engineered 3CLpro (L50F-E166A-L167F) nirmatrelvir resistant (3CLpro[res]) virus (passage 1 on Vero E6 cells) used here has been described before[18]. Live virus-related work was conducted in the high-containment A3 and BSL3 facilities of the KU Leuven Rega Institute (3CAPS) under licenses AMV 30112018 SBB 219 2018 0892 and AMV 23102017 SBB 219 20170589 according to institutional guidelines. Only authorized personnel that have received intensive training have strictly controlled access to these facilities.

### Cells
Vero E6 cells (African green monkey kidney, ATCC CRL-1586) were cultured in minimal essential medium (MEM, Gibco) supplemented with 10% fetal bovine serum (Integro), 1% non-essential amino acids (NEAA, Gibco), 1% L- glutamine (Gibco) and 1% bicarbonate (Gibco). End-point titrations on Vero E6 cells were performed with medium containing 2% fetal bovine serum instead of 10%. Cells were kept in a humidified 5% $CO_2$ incubator at 37 °C.

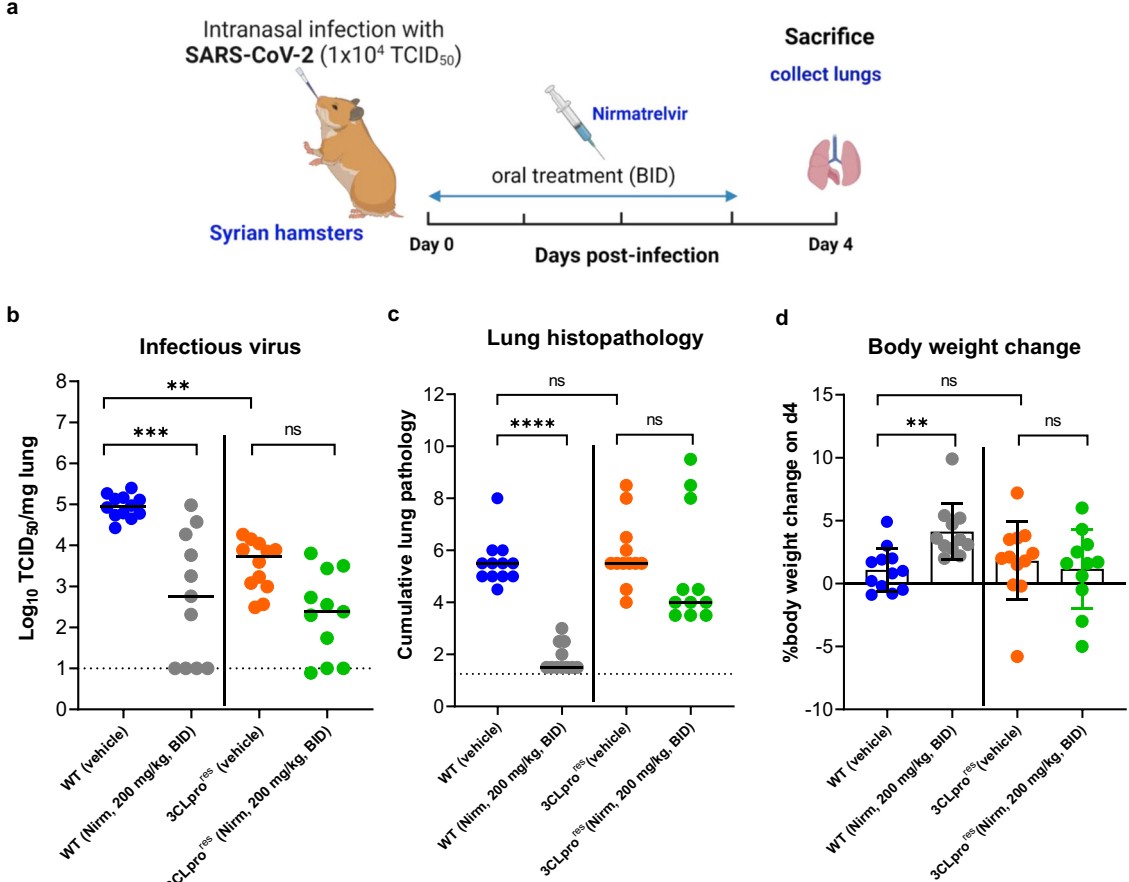

**Fig. 3 | In vivo efficacy of nirmatrelvir against the 3CLpro (L50F-E166A-L167F) resistant virus. a** Set-up of the study. **b** Infectious viral loads in the lungs of hamsters that were treated with vehicle or nirmatrelvir (Nirm) at 200 mg/kg (BID) and infected with $10^4$ TCID$_{50}$ of either the wild-type (WT) SARS-CoV-2 isolate (USA-WA1/2020) or the 3CLpro (L50F-E166A-L167F) nirmatrelvir resistant (3CLpro$^{res}$) virus at day 4 post-infection (pi) are expressed as log$_{10}$ TCID$_{50}$ per mg lung tissue. Individual data and median values are presented. **c** Cumulative severity score at day 4 p.i. from H&E stained slides of lungs from hamsters treated with either vehicle or nirmatrelvir (Nirm) and infected with either the WT SARS-CoV-2 isolate or 3CLpro$^{res}$ virus. Individual data and median values are presented; the dotted line represents the median score of untreated non-infected hamsters. **d** Weight change at day 4 pi in percentage, normalized to the body weight at the time of infection. Bars represent means ± SD. Data were analyzed with the Kruskal–Wallis test, **$p < 0.01$, ***$p < 0.001$, ****$p < 0.0001$, ns = non-significant. Data presented are from 2-independent studies with a total $n = 12$ per vehicle-treated groups and $n = 11$ for the nirmatrelvir-treated groups. Panel (**a**) designed by Biorender.

## Compound

Nirmatrelvir was synthesized at Excenen (China). In addition, a 1:1 MTBE solvate form of nirmatrelvir was kindly provided by Pfizer. The Excenen batch was used for the first experiment and the MTBE solvate form Pfizer was used for the second experiment. Nirmatrelvir (Excenen) was formulated as 100 mg/ml in a vehicle containing 43% ethanol, 27% propyele glycol (Sigma) in sterile distilled water. The MTBE solvate form (Pfizer) was formulated as 40 mg/ml suspension in 2% (v/v) Tween80 in 98% (v/v) of 0.5% (w/v) Methyl Cellulose by geometric dilution.

## Infection and transmission study

The hamster infection model of SARS-CoV-2 has been described before[23,24]. Female Syrian hamsters (Mesocricetus auratus) were purchased from Janvier Laboratories and kept per two in individually ventilated isolator cages (IsoCage N Bio-containment System, Tecniplast) at 21 °C, 55% humidity and 12:12 day/night cycles. Housing conditions and experimental procedures were approved by the ethics committee of animal experimentation of KU Leuven (license P065-2020). For infection, two groups of index female hamsters of 6-8 weeks old were anesthetized with ketamine/xylazine/atropine and inoculated intranasally with 50 μL containing 1×$10^4$ TCID$_{50}$ of either SARS-CoV-2 WT virus or the 3CLpro$^{res}$

virus(day 0). In the morning of day 1 post-infection (pi), each index hamster was co-housed with a contact (sentinel) hamster (non-infected hamsters) in one cage and the co-housing continued until the sacrifice day i.e. 3 days after start of exposure. On day 4 pi, animals were euthanized by intraperitoneal injection of 500 μL Dolethal (200 mg/mL sodium pentobarbital) and lungs were collected for further analysis. Two independent studies were performed each with a total of n = 12 per group.

## Nirmatrelvir efficacy study

Female Syrian hamsters were treated by oral gavage with either the vehicle or nirmatrelvir at 100 or 200 mg/kg/dose twice daily (BID) starting from D0, just before the infection with either SARS-CoV-2 WT virus or the 3CLprores virus (day 0) as described above. All the treatments continued until day 3 pi. Hamsters were monitored for appearance, behavior and weight. At day 4 pi, hamsters were euthanized by i.p. injection of 500 μL Dolethal (200 mg/mL sodium pentobarbital, Vétoquinol SA). Lungs were collected and viral RNA and infectious virus were quantified by RT-qPCR and end-point virus titration. Two independent studies were performed each with a total n = 12 for vehicle groups and n = 11 for 200 mg/kg nirmatrelvir groups. For 100 mg/kg dose group, a single experiment was performed with n = 6.

## SARS-CoV-2 RT-qPCR

Hamster lung tissues were collected after sacrifice and were homogenized using bead disruption (Precellys) in 350 μL TRK lysis buffer (E.Z.N.A.® Total RNA Kit, Omega Bio-tek) and centrifuged (10.000x g, 5 min) to pellet the cell debris. RNA was extracted according to the manufacturer's instructions. RT-qPCR was performed on a Light-Cycler96 platform (Roche) using the iTaq Universal Probes One-Step RT-qPCR kit (BioRad) with IDT nCOV_N2 primers and probes targeting the nucleocapsid (catalog # 10006824-26)[24]. Standards of SARS-CoV-2 cDNA (IDT) were used to express viral genome copies per mg tissue[23].

## End-point virus titrations

Lung tissues were homogenized using bead disruption (Precellys) in 350 μL minimal essential medium and centrifuged (10,000x g, 5 min, 4 °C) to pellet the cell debris. To quantify infectious SARS-CoV-2 particles, endpoint titrations were performed on confluent Vero E6 cells in 96- well plates. Viral titers were calculated by the Reed and Muench method[25] using the Lindenbach calculator and were expressed as 50% tissue culture infectious dose ($TCID_{50}$) per mg tissue.

## Histology

For histological examination, the lungs were fixed overnight in 4% formaldehyde and embedded in paraffin. Tissue sections (5 μm) were analyzed after staining with hematoxylin and eosin and scored blindly for lung damage by an expert pathologist. The scored parameters, to which a cumulative score of 1–3 was attributed, were the following: congestion, intra-alveolar hemorrhagic, apoptotic bodies in bronchus wall, necrotizing bronchiolitis, perivascular edema, bronchopneumonia, perivascular inflammation, peribronchial inflammation and vasculitis.

## Deep sequencing and analysis of whole genome sequences

The viral RNAs isolated from the lungs of infected animals in the transmission study and the nirmatrelvir-treated animals in the treatment study were subjected to deep sequencing analysis of the full genome sequence through ARTIC SARS-CoV-2 RNA-Seq service provided by Eurofins Genomics. Alignment of the obtained sequences was performed using the Geneious Prime software.

## Statistics

The detailed statistical comparisons, the number of animals and independent experiments that were performed is indicated in the legends to figures. "Independent experiments" means that experiments were repeated separately on different days. The analysis of histopathology was done blindly. All statistical analyses were performed using GraphPad Prism 9 software (GraphPad, San Diego, CA, USA). Statistical significance was determined using the non-parametric Mann Whitney U-test. P-values of <0.05 were considered significant.

## Ethics

Housing conditions and experimental procedures were done with the approval and under the guidelines of the ethics committee of animal experimentation of KU Leuven (license P065-2020).

## Reporting summary

Further information on research design is available in the Nature Portfolio Reporting Summary linked to this article.

## Data availability

All of the data generated or analysed during this study are included in this published article. The deep sequencing data of the viral RNA from the transmission study are available on SRA under BioProject: PRJNA925007 with accession numbers SRX19069060-SRX19069082 and SRX19058198. All data supporting the findings in this study are also available from the corresponding author upon request. All source data are provided as a Source Data file Source data are provided with this paper.

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

## Acknowledgements

We thank Carolien De Keyzer, Lindsey Bervoets, Thibault Francken, Stijn Hendrickx and Niels Cremers for excellent technical assistance. We thank Prof. Jef Arnout and Dr. Annelies Sterckx (KU Leuven Faculty of Medicine, Biomedical Sciences Group Management) and Animalia and Biosafety Departments of KU Leuven for facilitating the animal studies. We thank Dr. Rhonda D. Cardin (Pfizer) for providing us with the solvate form of nir-matrelvir. This project has received funding from the Covid-19-Fund KU Leuven/UZ Leuven and the COVID-19 call of FWO (G0G4820N), the European Union's Horizon 2020 research and innovation program under grant agreements No 101003627 (SCORE project). This work was also supported by the Belgian Federal Government for the VirusBank Platform.

## Author contributions

R.A. and J.N. designed the studies; R.A., K.D. and B.W. performed the studies and analyzed data; R.A. made the graphs; B.W., D.J. and J.N. provided advice on the interpretation of data; R.A. and J.N. wrote the paper; B.T., N.E. and V.T. provided the reverse-engineered virus; R.A., V.T. and J.N. supervised the study; D.J. and J.N. acquired funding.

## Competing interests

The authors declare no competing interests.
