## [Peer Review File · Nature Communications]

Nirmatrelvir-resistant SARS-CoV-2 is efficiently transmitted in Syrian hamsters and retains partial susceptibility to treatmentREVIEWER COMMENTS

Reviewer #1 (Remarks to the Author):

In the paper of Abdelnabi et al. there is an investigation of a nirmatrelvir-resistant mutant of SARS-CoV-2 on the WA1 background. The paper investigates whether the resistant virus can infect and transmit using the Syrian hamster model and find that the mutant virus is attenuated compared to wild-type but can transmit at a similar level. Even though the mutated virus is attenuated, the pathology in the hamster lungs is equivalent which raises questions about the immune response the "attenuated" virus is causing that is not investigated. It would make sense to investigate the immune response since the 3CLpro of coronaviruses is well established to play a role in immune antagonism. The paper then ends with another infection where a higher dose of nirmatrelvir is required to inhibit the resistant virus than seen for wild type, but that the resistant virus can still be inhibited by the drug treatment.

The data in the paper are somewhat interesting but are not substantial enough for publication being composed of just two infections of hamsters. The paper consistently references previous work performed to produce the resistant virus through passage in VeroE6 cells (reference 16, containing many of the same authors as this paper) which as far as I can ascertain is still only a bioRxiv pre-print. Were that data to be included with this in vivo data then the paper could be considered for publication, but the data presented here are not substantial enough for a standalone paper. Moreover, in figure 3 a "sub-optimal" dose of 200mg/kg nirmatrelvir is still inhibiting the passaged virus, raising the question of whether this is in fact a resistant virus or not. For instance, the virus could just be attenuated because of passage in VeroE6 cells, a better control would be an equivalently passaged virus, in the absence of nirmatrelvir for example. The authors may also wish to consider adding more discussion on the ethics and biocontainment around producing a SARS-CoV-2 virus that is resistant to a drug that is being used to treat patients that are being infected by circulating viruses.

Reviewer #2 (Remarks to the Author):

In this manuscript, Abdelnabi et al describe the in vivo infection of Syrian hamsters with their previously identified nirmatrelvir-resistant SARS-CoV-2 mutant in order to test the infectability, transmissibility, and nirmatrelvir inhibition of this mutant. While brief, this report is informative as the first in vivo analysis of a nirmatrelvir-resistant SARS-CoV-2 mutant. The authors demonstrate that the virus can readily infect Syrian hamsters and transmit to naïve hamsters, albeit at compromised levels compared to the WT. This corroborates this group's previous findings, as well as others in the literature, that the mutant virus has a fitness defect. Despite this defect, the mutant virus caused similar lung pathology as WT. Treatment of infected animals with nirmatrelvir showed that the mutant virus is resistant against a low dose, and moderately resistant against a high dose. The work is overall sound, but additional data would be helpful, especially for the treatment experiment, and this reviewer has some suggestions below for additions that should not be too much additional work yet would be meaningful.

Major comments:

- While the writing is overall clear, there are numerous spelling errors and some grammatical issues throughout. This reviewer has indicated the more obvious errors in the minor comments below, but likely has missed some others. While the need for expedience in SARS-CoV-2 research is understandable, a careful read and edit of the text is suggested.
- It is notable that the WT virus and the resistant virus differ greatly in viral titer (Fig 1) yet have no difference in lung pathology (Fig 2). Is there any difference in weight loss over time? Presumably such data are available as the authors show such data in their previous work (as an aside, this paper seems like it should be referenced in the main text/methods): <https://www.nature.com/articles/s41467-022-28354-0>. Some discussion on why there may be comparable lung pathology despite the difference seems warranted.
- Additional data for the nirmatrelvir treatment experiment would be helpful to put the 1.4 log₁₀ reduction in viral titer for the high dose against resistant virus in context, especially as it was not statistically significant. Weight loss data and lung histopathology scoring would perhaps be the most straightforward additions that could be added.
- The deep sequencing is appreciated to confirm that the mutations are retained in vivo. This reviewer suggests that this sequencing should be uploaded to a public repository since these data are not shown (such as to SRA). Were samples from the nirmatrelvir treated animals also sequenced? It would be quite simple yet informative to do so to understand if the surviving viruses in the WT group had gained resistance mutations, as well as whether those viruses in the mutant group gained additional mutations.
- As part of their previous study, this group examined the PK of nirmatrelvir in Syrian hamsters. Does the 51-fold loss in in vitro EC₅₀ of the mutant virus correlate with the treatment experiment here, when considering the previously determined plasma concentrations of nirmatrelvir in Syrian hamsters?

Minor comments:

- Line 20, 40, 51 – Ensitrelvir is now authorized in Japan, so perhaps it would be more accurate to note that this is referring to the US
- Line 22 – “yet” seems unneeded
- Line 26 – “Importanly” is spelled incorrectly
- Line 28 – “infecetd” is spelled incorrectly
- Line 41 – “analoges” is spelled incorrectly

- Line 50, 52 – it would be better to reference the original papers for the Paxlovid clinical trial and ensitrelvir drug development rather than these commentary pieces (refs 7, 8)
- Line 55, 57 – presumably the authors are referring to HIV-1 specifically
- Line 58 – “the clinical settings”, the “the” is unneeded
- Line 65 – “the 3CLpro”, the “the” is unneeded
- Line 66 – “aim” should be “aimed”
- Line 99 – “subsstitutions” is spelled incorrectly
- Line 115 – “recenty” is spelled incorrectly
- Line 116 – “ensetrilvir” is spelled incorrectly
- Line 121 – “subsstitutions” and “subsstitution” are spelled incorrectly
- Line 123 – “subsstitutions” is spelled incorrectly
- Line 166 – “amantadanes” is spelled incorrectly
- Line 243 – “Independednt” is spelled incorrectly
- Fig 2b – there seems to only be 11 datapoints for the resistant virus group, though the earlier figure and legend notes there were 12 animals.

January 31st 2023

Manuscript: NCOMMS-22-45423

LEUVEN

Response to Reviewers

Reviewer #1:

1. The data in the paper are somewhat interesting but are not substantial enough for publication being composed of just two infections of hamsters. The paper consistently references previous work performed to produce the resistant virus through passage in VeroE6 cells (reference 16, containing many of the same authors as this paper) which as far as I can ascertain is still only a bioRxiv pre-print. Were that data to be included with this in vivo data then the paper could be considered for publication, but the data presented here are not substantial enough for a standalone paper. In the manuscript to which the Reviewer refers, we report on the selection of SARS-CoV-2 variants that are resistant to various corona 3CLpro inhibitors and provide an in depth study of the particular characteristics of these strains and the molecular mechanism resulting in the resistant phenotype. This manuscript entitled "*The substitutions L50F, E166A and L167F in SARS-CoV-2 3CLpro are selected by a protease inhibitor in vitro and confer resistance to nirmatrelvir*" has meanwhile been published (Jochmans D *et al*, **mBio**. 2023 Jan 10:e0281522). We now refer to the mBio reference in our revised manuscript. We believe, with all due respect, that the data that we present in the current manuscript, is sufficiently substantiated for a standalone paper and that the message of the study is very relevant to the clinical context.

2. Moreover, in figure 3 a "sub-optimal" dose of 200mg/kg nirmatrelvir is still inhibiting the passaged virus, raising the question of whether this is in fact a resistant virus or not. For instance, the virus could just be attenuated because of passage in VeroE6 cells, a better control would be an equivalently passaged virus, in the absence of nirmatrelvir for example. The resistant virus used in the present study is [as was also mentioned on page 10: line 195-196 of the first version of the manuscript] the reverse-engineered virus that carries the L50F, E166A and L167F mutations and that was characterized in much detail (including the data on the drug-resistant phenotype) in our mBio paper (Jochmans D *et al*, **mBio**. 2023 Jan 10:e0281522). The virus used here is a passage 1 virus stock prepared on Vero E6 cells. Thus the virus is not passaged on or adapted to Vero E6.

In hamsters that had been intranasally infected with this resistant virus, a lung pathology was observed that was comparable to that in animals that had been infected with wild

type virus (Figure 2). Infection with either wild-type or resistant virus resulted in a comparable change in body weight as we now show in Supplementary figure S1. In hamsters infected with the drug-resistant virus, nirmatrelvir was less effective than in animals infected with the wild-type virus [at the same dose a reduction of respectively 1.4 \log_{10} infectious virus titers versus 2.2 \log_{10} in case of animals infected with the WT virus].

We now present in the revised manuscript also histopathology data for the treatment experiment (Fig 3C) and in the results section [page 6: lines 112-117]. We demonstrate that treatment with nirmatrelvir (200 mg/kg, BID) of hamsters infected with wild-type virus results in marked and significant reduction of lung pathology scores (reduction close to the baseline). In animals infected with the 3CLPro-resistant and that received the same dose of nirmatrelvir, only a minimal impact on virus-induced lung pathology is noted. This is another confirmation that the virus used is indeed resistant to 3CLpro inhibitors.

3. The authors may also wish to consider adding more discussion on the ethics and biocontainment around producing a SARS-CoV-2 virus that is resistant to a drug that is being used to treat patients that are being infected by circulating viruses. All live virus-related work was conducted in modern and state-of-the art high-containment A3 and BSL3 facilities of the KU Leuven Rega Institute (3CAPS) under licenses AMV 30112018 SBB 219 2018 0892 and AMV 23102017 SBB 219 20170589 according to institutional guidelines. Only authorized personnel that received intensive training have strictly controlled access to these facilities. This information was already mentioned in the method section of the original version [page 10: line 196-200].

Reviewer #2:

Major comments:

1. While the writing is overall clear, there are numerous spelling errors and some grammatical issues throughout. We apologize and carefully revised the language of the manuscript.

2. It is notable that the WT virus and the resistant virus differ greatly in viral titer (Fig 1) yet have no difference in lung pathology (Fig 2). Is there any difference in weight loss over time? Presumably such data are available as the authors show such data in their previous work (as an aside, this paper seems like it should be referenced in the main text/methods): <https://www.nature.com/articles/s41467-022-28354-0>. Some discussion on why there may be comparable lung pathology despite the difference seems warranted. The daily body weight change data have now been included as supplementary figure S1. As was the case for the lung histopathology data, no significant difference in %body weight change was observed between the WT and 3CLpro^{res} virus-infected groups throughout the infection period. In addition, only a slight difference (0.6 \log_{10}) was observed in the viral RNA levels in the lungs (Fig. 1B). The significant difference in the lung infectious virus titers between animals infected with the WT and the resistant virus could be a limitation of the end-point titration for infectious

virus (the resistant virus may have lower fitness on Vero E6 cells, which were used for the titration assay). We now comment on this [on page 7: lines 139-142].

3. Additional data for the nirmatrelvir treatment experiment would be helpful to put the 1.4 log₁₀ reduction in viral titer for the high dose against resistant virus in context, especially as it was not statistically significant. Weight loss data and lung histopathology scoring would perhaps be the most straightforward additions that could be added. We now include data on the lung histopathology data and %weight change in the revised manuscript (inserted in Figure 3 as panels C and D), respectively and described the findings in the Results section [page 6: lines 112-120]. Indeed, a clear shift in antiviral efficacy was observed when assessing the effect of the drug on the lung histopathology scores. Upon treatment with nirmatrelvir, a marked and significant reduction of lung histopathology scores (close the baseline score observed in non-infected untreated hamsters) was only observed in the animals infected with the WT virus (Fig. 3C). In animals infected with the 3CLpro^{res} virus, treatment with nirmatrelvir resulted only in slight reduction of the lung histopathology scores (median score of 4.25) as compared to the corresponding vehicle treated group (median score of 5.5), (Fig. 3C). No significant weight loss was observed in either the vehicle or nirmatrelvir-treated groups on day 4 post-infection (Fig. 3D). However, animals infected with the WT virus and that had received nirmatrelvir had a significantly higher weight gain (mean % weight change of 4.1) as compared to the other groups (mean % weight change <2) (Fig. 3D).

4. The deep sequencing is appreciated to confirm that the mutations are retained in vivo. The reviewer suggests that this sequencing should be uploaded to a public repository since these data are not shown (such as to SRA). The deep sequencing data of the transmission study have been uploaded to SRA under BioProject: PRJNA925007 with accession numbers from SRX19069060-SRX19069082 and SRX19058198. We have included this statement under Data availability section [page 13: lines 273-275].

5. Were samples from the nirmatrelvir treated animals also sequenced? It would be quite simple yet informative to do so to understand if the surviving viruses in the WT group had gained resistance mutations, as well as whether those viruses in the mutant group gained additional mutations. Indeed, we deep sequenced the viral RNA from the lungs of nirmatrelvir-treated animals infected with either the WT virus or the 3CLpro^{res} virus. The sequencing data revealed that there are no changes in the 3CLpro gene of the WT virus as well as no additional mutations in the 3CLpro of the resistant variant. We included the sequence alignment as supplementary figure S2.

6. As part of their previous study, this group examined the PK of nirmatrelvir in Syrian hamsters. Does the 51-fold loss in in vitro EC₅₀ of the mutant virus correlate with the treatment experiment here, when considering the previously determined plasma concentrations of nirmatrelvir in Syrian hamsters? In our earlier published study, we used in hamsters a slightly higher dose of nirmatrelvir (i.e. 250 mg/kg instead of 200 mg/kg in the present study, Abdelnabi R et al. *Nat Commun* 2022; 13, 719). This dose resulted in peak plasma levels of 22.4 ± 11 μM and peak lung exposure of 6.2 ± 4.9 μM (at 3 hours post exposure). In Vero E6, we observed an EC₅₀ of nirmatrelvir of 6.1 μM

against the resistant variant. With the 200 mg/kg dose (used in our current study), we expect to have lung exposure somewhat below 6 μM . This may explain the residual (but not efficient) activity of the 200 mg/kg dose against the resistant virus in our efficacy experiment. We have included this information now in the discussion section [page 8: lines 160-166].

Minor comments:

- **Line 20, 40, 51 – Ensitrelvir is now authorized in Japan, so perhaps it would be more accurate to note that this is referring to the US.** Thank you. We rephrased the sentences (now at line 19 and 41) by referring to the FDA and EMA and to the approval of ensitrelvir in Japan (now in line 52).
- **Line 22 – “yet” seems unneeded.** The word was removed.
- **Line 26 – “Importantly” is spelled incorrectly.** This was corrected.
- **Line 28 – “infecetd” is spelled incorrectly.** This was corrected.
- **Line 41 – “analoges” is spelled incorrectly.** This was corrected.
- **Line 50, 52 – it would be better to reference the original papers for the Paxlovid clinical trial and ensitrelvir drug development rather than these commentary pieces (refs 7, 8).** Thank you. We now use the original reference for the paxlovid clinical trial (Hammond J, et al. N. Engl. J. Med. 2022; 386, 1397–1408) and the reference for ensitrelvir (Unoh Y, et al. J Med Chem. 2022;65(9):6499-6512).
- **Line 55, 57 – presumably the authors are referring to HIV-1 specifically.** We rephrased the sentence and included references also for drug-resistance of herpes viruses and HBV.
- **Line 58 – “the clinical settings”, the “the” is unneeded.** This was corrected.
- **Line 65 – “the 3CLpro”, the “the” is unneeded.** This was corrected.
- **Line 66 – “aim” should be “aimed”.** This was corrected.
- **Line 99 – “subsstitutions” is spelled incorrectly.** This was corrected.
- **Line 115 – “recenty” is spelled incorrectly.** This was corrected.
- **Line 116 – “ensetrilvir” is spelled incorrectly.** This was corrected.
- **Line 121 – “subsstitutions” and “subsstitution” are spelled incorrectly.** This was corrected.
- **Line 123 – “subsstitutions” is spelled incorrectly.** This was corrected.
- **Line 166 – “amantadanes” is spelled incorrectly.** This was corrected.
- **Line 243 – “Independednt” is spelled incorrectly.** This was corrected.
- **Fig 2b – there seems to only be 11 datapoints for the resistant virus group, though the earlier figure and legend notes there were 12 animals.** Indeed, there was one point out of the axis range (score of 9). We have adjusted the Y-axis to show this point.

REVIEWERS' COMMENTS

Reviewer #2 (Remarks to the Author):

The authors have largely addressed the concerns of this reviewer in this revision, although I have a few points to raise for consideration before publication:

- The authors suggest that the resistant virus may have lower fitness on Vero E6 cells and that this may explain why a difference in lung infectious virus was observed, despite there being no difference in lung pathology. Perhaps this is the case, although this seems unexpected given that the authors show in their companion piece in mBio that the enzymatic activity of the 3CL protease with these mutations is reduced, suggesting that it is a fitness defect intrinsic to this mutant virus and not one dependent on the cell line used. It would be quite simple to conduct the infectious virus titration and fitness assay with the two viruses in a different cell line to validate the authors' hypothesis, which seems particularly appropriate given most studies investigating nirmatrelvir utilize other cells because of the issue of nirmatrelvir being a P-gp substrate, and it would be suggested for this high-quality Journal.
- In the additional information that the authors have added in this revision, they note that the WT virus has been passaged five times in Vero cells whereas the mutant virus is passage 1. It would be apt for the authors to confirm and state in the manuscript that the WT virus has not gained any additional mutations elsewhere in the genome through this passaging given that SARS-CoV-2 adaptation to Vero cells has been well-documented (e.g., through mutations in the spike), and this would affect their experiments and data interpretations. The authors note in their methods that they conducted whole genome sequencing so this data should already be available.
- The authors seem to have removed the 100 mg/kg BID group data from Figure 3 in the revision, but this reviewer suggests that the data should be retained as it nicely showed a dose-dependent response to the treatment in this model. Perhaps it can be incorporated as a supplement if it does not fit into Figure 3.
- Line 295 has some typos.

February 27th 2023

Manuscript: NCOMMS-22-45423

LEUVEN

Response to Reviewers

Reviewer #2:

• The authors suggest that the resistant virus may have lower fitness on Vero E6 cells and that this may explain why a difference in lung infectious virus was observed, despite there being no difference in lung pathology. Perhaps this is the case, although this seems unexpected given that the authors show in their companion piece in mBio that the enzymatic activity of the 3CL protease with these mutations is reduced, suggesting that it is a fitness defect intrinsic to this mutant virus and not one dependent on the cell line used. It would be quite simple to conduct the infectious virus titration and fitness assay with the two viruses in a different cell line to validate the authors' hypothesis, which seems particularly appropriate given most studies investigating nirmatrelvir utilize other cells because of the issue of nirmatrelvir being a P-gp substrate, and it would be suggested for this high-quality Journal. We have performed titration of the virus stocks of both WT and resistant variants in A549-ACEII-TMPRSS and Vero E6 cells in parallel. For the WT virus, the TCID₅₀/mL on Vero E6 and A549 duals were 8.9E+05 and 1.5E+06, respectively. On the other hand, the TCID₅₀/mL of the resistant virus on Vero E6 and A549 duals were 2.8E+05 and 8.9E+07, respectively. We can as yet not explain this difference between the two cell lines as there are no major differences in the spike genes of both SARS-CoV-2 stocks as we will explain in the next comment and as we now present in supplementary Figure S4. Since we have no clear hypothesis, we have inserted the following in the Discussion section "*Intranasal infection of female Syrian hamsters with either the WT or the 3CLpro^{res} virus resulted in efficient replication of the virus in the lungs. However, the viral RNA loads and infectious titers in the lungs of hamsters infected with the 3CLpro^{res} virus were 0.6 and 1.9 log₁₀ lower compared to the WT virus-infected index hamsters. The relative lower infectious virus titers as compared to the RNA levels of the 3CLpro^{res} virus (isolated from the lungs of hamsters) seems to indicate that this 3CLpro^{res} virus has a lower relative infectivity (ratio viral RNA/infectious titers) than the WT virus. We have as yet no explanation of this observation.*"

• In the additional information that the authors have added in this revision, they note that the WT virus has been passaged five times in Vero cells whereas the mutant virus is passage 1. It would be apt for the authors to confirm and state in the manuscript that the WT virus has not gained any additional mutations elsewhere in

the genome through this passaging given that SARS-CoV-2 adaptation to Vero cells has been well-documented (e.g., through mutations in the spike), and this would affect their experiments and data interpretations. The authors note in their methods that they conducted whole genome sequencing so this data should already be available. We have included the spike gene sequence data as supplementary Figure S4. In fact, there are some minor variations in the spike genes of the WT and the 3CLpro^{res} virus inocula used for infection (no Vero adaptation mutations or deletions were detected). Passaging the WT virus through the hamsters seems to select for several mutations in the S-protein which is not the case for the 3CLpro resistant virus. It might be speculated that the higher replication capacity of the WT virus *versus* the 3CLpro^{res} in the hamster may cause more selection pressure on the spike. We did not further investigate this as it is out of the scope of this study .

- **The authors seem to have removed the 100 mg/kg BID group data from Figure 3 in the revision, but this reviewer suggests that the data should be retained as it nicely showed a dose-dependent response to the treatment in this model. Perhaps it can be incorporated as a supplement if it does not fit into Figure 3.** We now inserted again the dataset on the 100 mg/kg as supplementary Figure S2.
- **Line 295 has some typos.** We have corrected the typos.